# Enhancing Capsid Proteins Capacity in Plant Virus-Vector Interactions and Virus Transmission

**DOI:** 10.3390/cells10010090

**Published:** 2021-01-07

**Authors:** Alexey Agranovsky

**Affiliations:** Biological Faculty, Moscow State University, 119234 Moscow, Russia; aaa@genebee.msu.su

**Keywords:** plant RNA viruses, luteoviruses, benyviruses, closteroviruses, capsid protein, readthrough domain, minor capsid protein, vector transmission

## Abstract

Vector transmission of plant viruses is basically of two types that depend on the virus helper component proteins or the capsid proteins. A number of plant viruses belonging to disparate groups have developed unusual capsid proteins providing for interactions with the vector. Thus, cauliflower mosaic virus, a plant pararetrovirus, employs a virion associated p3 protein, the major capsid protein, and a helper component for the semi-persistent transmission by aphids. Benyviruses encode a capsid protein readthrough domain (CP-RTD) located at one end of the rod-like helical particle, which serves for the virus transmission by soil fungal zoospores. Likewise, the CP-RTD, being a minor component of the luteovirus icosahedral virions, provides for persistent, circulative aphid transmission. Closteroviruses encode several CPs and virion-associated proteins that form the filamentous helical particles and mediate transmission by aphid, whitefly, or mealybug vectors. The variable strategies of transmission and evolutionary ‘inventions’ of the unusual capsid proteins of plant RNA viruses are discussed.

## 1. Introduction

Transmission of plant viruses is a complex virus–vector–host interplay tuned in the course of evolution. This process is basically of two types, one specified by the requirement for a virus helper component protein (HC), which serves as an ‘adaptor’ between the virus capsid protein (CP) and a receptor in the vector, and the other by the direct interaction of the assembled CP with a vector receptor [1,2,3]. The helper strategy was first discovered and studied in depth for aphid transmission of cauliflower mosaic virus (CaMV) [4,5,6,7,8] and the members of the *Potyviridae* family [9,10]. A classic example of the capsid strategy is cucumber mosaic virus (CMV, family *Bromoviridae*), a small icosahedral virus composed of 180 identical CP subunits whose aphid transmission depends solely on the capsid protein [11,12].

In the process of evolution, a number of plant viruses have developed peculiar capsid proteins providing for interactions with the vector. Thus, CaMV, apart from the nonvirion helper component, employs the capsid protein p3 for semi-persistent aphid transmission. The capsid protein readthrough domain (CP-RTD), being a minor component of icosahedral virions, mediates persistent, circulative aphid transmission of luteoviruses. Likewise, benyviruses encode the CP-RTD exposed at one end of the rod-like helical particle, which serves for the virus transmission by soil fungal zoospores. Closteroviruses encode several CPs and virion-associated proteins that form a complex structure of the filamentous helical particles and provide for the transmission by aphid, whitefly, or mealybug vectors. This review focuses on variable transmission strategies and evolutionary ‘inventions’ of the unusual capsid proteins of plant viruses. An insight into the transmission mechanisms of these viruses is not only of fundamental but also of practical interest in view of significant economic losses from diseases such as beet rhizomania [13], cereal yellow dwarf [14,15], beet yellows, and citrus tristeza [16].

## 2. The Capsid Proteins in Semi-Persistent Aphid Transmission of Caulimoviruses

Cauliflower mosaic virus, a plant pararetrovirus, has a circular 8-kbp DNA genome and icosahedral particles (approximately 50 nm in diameter) built of 420 CP subunits [17]. After the virus enters the cell, the DNA is delivered into the nucleus and transcribed into the capped and polyadenylated genome-length 35S RNA and the 19S subgenomic RNA for transactivator protein/viroplasmine (TAV). Translation of polycistronic 35S mRNA occurs via ribosomal shunting and re-initiation powered by TAV and its cellular partner proteins to yield the proteins for replication, packaging, aphid transmission and other accessory functions [18] (Figure 1).

CaMV is semi-persistently transmitted by several aphid species of which *Brevicoryne brassicae* and *Myzus persicae* are the major field vectors [19]. The virus is acquired by aphids from epidermal and mesophyll cells during brief punctures; the acquisition significantly increases when the vector’s stylet reaches the phloem, and the subsequent inoculation occurs via intracellular salivation upon stylet punctures of the epidermal and mesophyll tissues [19]. CaMV particles are retained at the extreme tip of the maxillary stylet, where they likely interact, via the HC, with an aphid protein Stylin-1, which is thus a likely candidate to be a transmission receptor [20,21]. The key viral proteins involved in transmission, are the minor CP (gene III product, p3, referred to here as CPm in keeping with protein designations in other sections of this review) and the nonvirion helper component (HC, or p2, gene II product) [5,6,7] (Figure 1). CaMV acquisition depends on the ‘transmission body’ in the infected plant cells, a dynamic ultrastructure accumulating the HC [22]. The CaMV CPm is inserted into the virion between the CP capsomers, with the CPm C-terminus embedded in the inner shell and the N-terminus exposed to the solution. In virions, the CPm forms homodimers via coiled-coil interactions of the N-terminal domains [7]. The N-terminal domain of the virion-associated CPm drives interactions with the CaMV HC [23]. Oligomerization status may serve as a switch between the distinct functions of the CaMV CPm in vector transmission, cell-to-cell movement, and long distance transport, and may also determine its contacts with partner proteins [7].

## 3. CP-RTD Domain in Persistent, Circulative Aphid Transmission of Luteoviruses

Luteoviruses cause phloem-limited infections in cereals, which result in significant crop losses worldwide [14,15,24]. They have small nonenveloped capsids (about 25 nm in diameter) that encompass monopartite positive sense RNA genome of 5.5–6 kb [25]. Genomic RNA lacks a 5′-cap structure (some, but not all, luteoviruses bear a 5′-genome linked protein) and has no 3′-poly(A). The genome contains five to six ORFs (Figure 2) [25]. Prior to translation, luteovirus RNA adopts a circular conformation owing to the complementary sequences in the 5′- and 3′-noncoding regions, and the translation initiation is mediated by the 3′-cap-independent translational enhancer, a signal that attracts the 40S ribosomal subunit and initiation factors [26,27,28]. The genome expression pattern includes a number of non-orthodox translation mechanisms, i.e., −1 frameshifting for the RNA-dependent RNA polymerase (Pol), leaky scanning for the CP and vascular transport protein (TP), and suppression of the leaky stop codon for the CP fused to the C-terminal readthrough domain (CP-RTD) [25] (Figure 2). The elaborate readthrough signal in BYDV includes a cytosine-rich sequence 3′ to the UGA stop codon and a complementary sequence element located approximately 700 nt downstream [25]. The fusion protein is included in luteovirus capsids as a minor component; conceivably, the readthrough mechanism serves to downregulate the CP-RTD synthesis (Figure 2) [29,30].

Luteoviruses are transmitted by aphids in a persistent, circulative mode, with acquisition and inoculation access times of many hours and virus retention throughout life of the vector. The virus-vector interactions are highly specific, with each luteovirus transmitted by one or a few aphid species [25,31]. The barley yellow dwarf viruses are traditionally named after their aphid vectors, e.g., BYDV RPV (currently renamed as cereal yellow dwarf virus, CYDV RPV, transmitted by *Rhopalosiphum padi*), BYDV MAV (*Macrosiphum* (*Sitobion*) *avenae*), BYDV RMV (*Rhopasosiphum maidis*) [31].

Gray and Gildow [31] recognized four distinct stages in luteovirus transmission: (a) Acquisition from the phloem of infected plant into the aphid’s alimentary canal; (b) passage through the gut; (c) retention in the hemocoel; and (d) transfer into the salivary gland and transmission into the phloem cells of a host. In this circulative route, luteoviruses have to cross the membranes of the hindgut epithelium, the basal lamina and the salivary gland plasmalemma [25,31], which is apparently achieved by receptor-mediated endocytosis [31,32]. According to the in vitro data, the alanyl aminopeptidase N, a membrane-associated protein of pea aphid, *Acyrthosiphon pisum*, directly binds to the major CP of pea enation mosaic enamovirus and may be a receptor for the virus [33]. Several aphid proteins (e.g., actin, glyceraldehyde 3-phosphate dehydrogenase, and Rack-1) were reported to bind to luteovirus particles, but the significance of these interactions in virus transmission is as yet unclear [34,35]. An abundant protein called symbionin, a GroEL homolog produced in aphids by endosymbiotic *Buchnera* spp., binds luteovirus particles and influences their stability and transmission [36], although its direct involvement in virus transmission is questionable [31].

On the virus side, the CP-RTD appears to be the key transmission receptor. Luteovirus CP-RTD is a multifunctional protein containing the major CP domain, a proline-rich spacer between the CP and RTD sequences, the conserved N-terminal and variable C-terminal regions of the RTD [25,37]. The RTD C-terminal region is dispensable for transmission but is involved in luteovirus long-distance transport, accumulation, tissue tropism, and symptoms expression [38,39], whereas the N-terminus of RTD directs incorporation of the fusion protein in virions, interactions with the symbionin, and aphid transmission [36,37,40]. Noteworthy, the closely spaced point mutations in the beet western yellows virus RTD N-terminal region had effects on distinct stages of the virus circulative pathway, the gut transfer and the hemocoel–salivary gland transfer [40]. The deleterious point mutations in the RTD were rescued in the progeny by the second site (compensatory) mutations rather than true reversions, suggesting a structural redundancy of the RTD [40].

## 4. CP-RTD in Fungal Transmission of Benyviruses

Beet necrotic yellow vein virus (BNYVV, family *Benyviridae*) is the causal agent of rhyzomania, a devastating and widely distributed disease of sugar beet [13,41]. The virus has rod-like helical symmetry particles. The BNYVV genome is divided among four positive-sense genomic RNAs (RNAs 1–4) that are capped and polyadenylated [42,43]. RNA-1 codes for the replication-associated polyprotein with the conserved domains of methyltransferase (Mtr), helicase (Hel), papain-like cysteine proteinase (PCP), and RNA dependent RNA polymerase (Pol) (Figure 3). RNA-2 encompasses the genes for CP-RTD, the Triple Gene Block (TGB) of cell-to-cell movement proteins, and the gene for a small cysteine rich protein. Monocistronic RNA-3 and RNA-4 code for the accessory nonstructural proteins p25 and p31 (Figure 3). Some BNYVV isolates contain an additional RNA component (RNA-5) influencing the severity of symptoms [42,43].

While the translation of monocistronic RNA-1, RNA-3, and RNA-4 is apparently straightforward, the expression of RNA-2 occurs via the readthrough mechanism and production of subgenomic RNAs [42,43]. The UAG stop codon in the CP gene is suppressed by ribosomes with ~10% efficiency to give rise to the CP-RTD fusion (Figure 3). It is possible that the BNYVV readthrough signal includes a downstream secondary structure, as is the case with luteoviruses [44].

BNYVV and several other viruses associated with rhyzomania (beet soil-borne virus, beet soil-borne mosaic virus, and beet virus Q) are transmitted by zoospores of soil fungus *Polymyxa betae*, an obligate parasite associated with cortical and epidermal root cells [45,46]. This fungus has a world-wide distribution [13]. Once the virus is acquired by *Polymyxa* zoospores, it is retained through different stages of the fungus development, including sporosores (the resting spores), which explains the benyvirus survival in the infested soil for years [47,48]. The BNYVV nonstructural proteins have been detected in *P. betae*, but the question as to whether the virus replicates in the vector remains unclear [48]. 

BNYVV particles are transmitted by the viruliferous zoospores upon fusion of the vector and the host cytoplasm [47]. This is not a merely passive process, since the transmission is mediated by the benyvirus proteins. The CP-RTD, being exposed in one or a few copies at one end of the BNYVV particles [49], specifically influences the virus transmission [50]. Deletions within the RTD and point mutations of the KTER domain (amino acids 553 to 556 of the RTD) block the fungal transmission [51]. Mutations in the RNA-4 gene also have a detrimental effect on the vector transmission, suggesting that the nonstructural p31 protein assists the BNYVV acquisition and/or inoculation by *P. betae* zoospores [52,53]. It remains to be determined if the p31 is a bona fide helper component of BNYVV, or its action in the transmission is indirect.

## 5. Unusual Capsid Proteins in Aphid, Whitefly, and Mealybug Transmission of Closteroviruses

The family *Closteroviridae* includes about 40 plant positive-sense RNA viruses characterized by several distinct features: (1) Semi-persistent mode of insect transmission (with the acquisition and inoculation access times of at least 30 min and the retention of a virus in its vector for up to 72 h); (2) phloem-limited or semi-phloem limited mode of plant infection (in the latter case, a virus primarily accumulates in the phloem but also invades the companion cells and mesophyll); (3) unique ‘rattlesnake-like’ structure of flexuous helical symmetry particles built of four proteins; (4) large RNA genomes (up to 20 kb); (5) the presence of the gene for a homolog of cell HSP70 chaperones and the duplicated genes or gene fragments (e.g., those for the major and minor CPs) [54,55,56]. The genomes are monopartite (genera *Closterovirus*, *Ampelovirus*, and *Velarivirus*) or divided among two or three genomic RNA segments (genus *Crinivirus*) (Figure 4).

Closterovirus RNA genomes are monopartite, with an exception of *Crinivirus* members that have the divided genomes (Figure 4). Genomic RNAs contains the 5′-cap and lacks a 3′-poly(A) tail [55,57,58]. The 5′-terminal portion encompasses overlapping ORFs 1a and 1b coding for the replication-associated proteins with the conserved PCP, Mtr, Hel, and Pol domains (Figure 4). Translation of these genes involving +1 ribosomal frameshifting results in the 1a and 1ab polyproteins. The beet yellows virus (BYV) PCP autocatalytically releases the leader protein [58]. The PCP domain in the genomes of some other closteroviruses is duplicated, and two leader proteins are released after cleavage [59]. The leader PCP of BYV influences the amplification of the virus RNA and the long-distance transport of virus infection through the host conductive tissues [60,61]. In addition to the cleavage by the PCP, the 1a polyprotein undergoes additional processing by an as yet unknown proteolytic activity, which yields the replication-associated proteins of 63 kDa (Mtr) and 100 kDa (HEL) [62] (Figure 4).

The 3′-proximal genes for structural and accessory proteins are expressed via a set of 3′-coterminal subgenomic RNAs [55,56] (Figure 4). The homolog of cell heat shock 70 kDa proteins (HSP70h), the ca. 60 kDa protein (p60), the minor capsid protein (CPm), and the major CP form the complex structure of flexuous filamentous particles of closteroviruses which is related to vector transmission [55]. Noteworthy, the BYV CP, CPm, and p60, despite significant divergence, have retained the profile of conserved amino acid residues characteristic of the CPs of helical symmetry viruses [63,64,65]. BYV HSP70h has a conserved N-terminal ATPase domain (homologous to the equivalent domains of the cell HSP70s and possessing the ATPase activity), and a variable C-terminal domain [66,67]. The cell-to-cell movement of BYV infection is mediated by the p6, HSP70h, p60, CPm, and CP; the products of the 3′-most genes are involved in long distance transport of the infection (p20) and suppression of post-transcriptional gene silencing (p21) [56] (Figure 4).

In closterovirus particles, the CP coats approximately 95% of the genomic RNA thereby forming the ‘body’ of filamentous particle, while the CPm forms a ‘tail’ that includes the 5′-terminal portion of the genome [68,69,70,71]. The assembly of BYV particles also requires HSP70h and p60, and both proteins, in one or a few copies, are associated with the mature particles [65,72]. The ‘rattlesnake-like’ particle structure is a hallmark of the *Closteroviridae* family [56,68].

Closteroviruses are semi-persistently transmitted by aphids (members of the genus *Closterovirus*), whiteflies (genus *Crinivirus*), or mealybugs (genus *Ampelovirus*). Transmission of lettuce infectious yellows virus (LIYV) by *Bemisia tabaci* New World has been studied in considerable depth with respect to vector retention sites and virus determinants of transmission [2]. Chen et al. [73] used sequential feeding of vector and non-vector biotypes of *B. tabaci* on artificial diets with LIYV particles or recombinant capsid proteins, followed by diets with fluorescently labeled antibodies to visualize the virus and the CPs in the vector by fluorescence microscopy and confocal laser scanning microscopy. It was found that LIYV associated with the anterior foregut (cibarium) of the vector biotype A, but not of the non-vector biotype B (MEAM1) [73]. Whiteflies retained more recombinant CPm compared to CP, p60, and HSP70h when fed on diets with each of the LIYV capsid proteins [73]. This is consistent with the fact that pre-incubation of the LIYV particles with CPm antibodies, but not the antibodies specific to the CP, p60, or HSP70h, blocked the virus uptake by *B. tabaci* [71]. A frameshift mutation that led to expression of only the N-terminal half of the LIYV CPm resulted in assembly of the truncated CPm into the virus particles and their successful long-distance transport in a host plant, but not the whitefly transmission of the mutant [74]. It should be noted parenthetically that this result is not easily reconciled with the protein sequence alignments, which imply that the C-terminal part of the LIYV CPm contains a conserved region that is likely to interact with the virus RNA [65,75]. Nevertheless, these lines of evidence indicate that the CPm is the key determinant of crinivirus whitefly transmission (Figure 4). The absence of inhibitory effects of antibodies to HSP70h and p60 does not allow excluding the influence of these proteins on the LIYV uptake and transmission. Unlike the CP and CPm, the HSP70h and p60 have not been visualized in the closteroviruses particles by immunospecific electron microscopy [71], possibly because their specific antigenic status in the particle is not compatible with polyclonal antibodies against the recombinant proteins used in the transmission neutralization assays. 

Members of the *Closterovirus* genus are semi-persistently transmitted by aphids. The mode and parameters of BYV transmission by green peach aphid *Myzus persicae* were described in early studies [76,77]. The vector specificity of closteroviruses is broader than that of criniviruses. Among 24 aphid species capable of transmitting BYV to sugar beets, *Myzus pesicae* and *Aphis fabae* are the most efficient and are the principal vectors in the field [77]. Likewise, among seven aphid species transmitting citrus tristeza virus (CTV), the brown citrus aphid *Toxoptera citricida* is the most efficient vector [78]. Transmission of BYV depends on a specific puncturing pattern of *M. persicae* detected by electrical penetration graph technique, where the vector’s stylet penetrates the membrane of phloem cells prior to the phloem salivation [79]. Killiny et al. [80] used the GFP-tagged CPm to monitor the CTV interactions with *Toxoptera citricida*. The fluorescent CTV particles were associated with the lining of the aphid’s cibarium, and this specific binding was reduced by competition with extraneously added CPm, HSP70h, or p60, but not with any other CTV-encoded protein (PCP1, PCP2, CP, p23, p20, p18, or p13) [80]. Pre-treatment of thin sections of the aphid’s stylet-cibarium-foregut structure with proteinase E, proteinase K, and trypsin had no effect on the specific binding of the fluorescent CTV, whereas pre-treatment with chitinase decreased the binding; in line with this, the binding dropped in competition of fluorescent CTV particles with glucose, D-glucosamine, chitobiose, and chitotriose. These data suggest that the closterovirus particles are retained in the cibarium of the aphid via specific interactions with sugar moieties [80]. 

Members of the genus *Ampelovirus* are semi-persistently transmitted by mealybugs (*Pseudococcidae*) and soft scale insects (*Coccidae*) [81,82,83,84]. The specificity of virus-vector interactions is limited; for example, a single species, *Pseudococcus longispinus* or *Phenacoccus aceris*, is capable of transmitting several individual viruses in the grapevine leafroll-associated virus complex [84,85], and four mealybug species transmit grapevine leafroll associated virus-3 (GLRaV-3) [86]. The data on the GLRaV-3 localization in mealybugs are somewhat contradictory, as Cid et al. [87] detected the virus by RT-PCR and immunogold labeling in primary salivary glands of *Planococcus citri*, whereas Prator et al. [88] observed the retention of the fluorescently labeled GLRaV-3 in the anterior foregut (cibarium) and retracted stylet tips of the *P. ficus*. The virus binding was blocked in competition with the wheat germ agglutinin added to the mealybug artificial diet, thus providing further analogy with the CTV transmission pattern [88]. The virus-encoded determinants of the mealybug transmission are as yet unknown.

## 6. Evolution of Capsid Proteins as Virus Transmission Determinants

The rapid evolution of viral RNA genomes, owing to the low fidelity of RNA-dependent RNA polymerases [89,90] and RNA recombination [91,92], has led to acquisition and modification of genes needed for virus adaptation. The reverse transcriptase-based replication of plant pararetroviruses is prone to recombination and is believed to be more accurate than RNA replication by at least one order of magnitude, yet the resulting evolution rates of CaMV are comparable with those of RNA viruses [93]. The new virus genes are likely to arise from heterologous recombination, from gene duplication and (asymmetric) divergence, and from de novo translation on an existing gene in a different reading frame (“overprinting”) [92,94,95]. While most of such sequences are destined to become extinct, some of them survive and provide an advantage to the virus. On the other hand, the high mutation rates and packaging constraints limit the expansion of size and coding capacity of RNA genomes [96,97]. The majority of RNA viruses have compact genomes of 4 to 12 kb which is largely due to these restricting factors [98].

These evolutionary trends concern the capsid proteins influencing the plant RNA virus-vector interactions and vector transmission. Plant pararetroviruses have acquired a unique duet of proteins, the p3 CPm and p2 helper component, that mediate the virus attachment to a receptor in the aphid vector. Among RNA viruses, members of the families *Luteoviridae* and *Benyviridae* have adapted the unusual capsid protein CP-RTD clinging the virus particle to vector receptor(s) to ensure its retention and transmission. The TP-CP-RTD gene module of luteoviruses and the CP-RTD of benyviruses are likely to have arisen by gene overprinting, gene duplication, and divergence of coding sequences [94]. The *Closteroviridae* have followed a distinct evolutionary path that has led to acquisition of more complex (and, possibly, more accurate) RNA replicase and super flexible capsids, which has allowed the expansion of their RNA genomes up to 19 kb [55,97]. The closterovirus determinants of vector transmission are the minor capsid proteins CPm, HSP70h, and p60. While the HSP70h gene has been possibly acquired by recombination between the RNA genome of a closterovirus progenitor with a cell mRNA for HSP70 chaperone [66], the CPm and p60 genes have most likely resulted from the major CP gene duplication and divergence [54,55,56]. Closterovirus genomes display traces of numerous gene duplication events, with some family members containing the duplicated leader proteinases and two diverged CPm genes (reviewed in [55]). Closterovirus particle consists of the CP-coated body and the CPm-coated tail, with the HSP70h and p60 possibly residing at the junction of these parts [55]. Apparently, this complex tail structure mediates the interaction of closteroviruses with their vectors to ensure semi-persistent transmission. The capsid proteins of closteroviruses interact with sugar moieties in the alimentary canal of their vectors, which is reminiscent of the receptors in animal virus hosts and vectors [99,100].

Despite some progress in studying the transmission determinants of caulimoviruses, luteoviruses, benyviruses, and closteroviruses, many gaps remain in our knowledge, particularly concerning the spatial structure of virus proteins mediating vector transmission and the mechanisms of their interaction with the vector receptors. Further studies are expected to give insights into the co-evolution of the virus–vector relationships and their molecular biological nature, thereby providing a basis for the development of measures to control virus transmission and reduce crop losses from the diseases such as rhyzomania, citrus tristeza, and cereal yellow dwarf.

## Figures and Tables

**Figure 1 cells-10-00090-f001:**
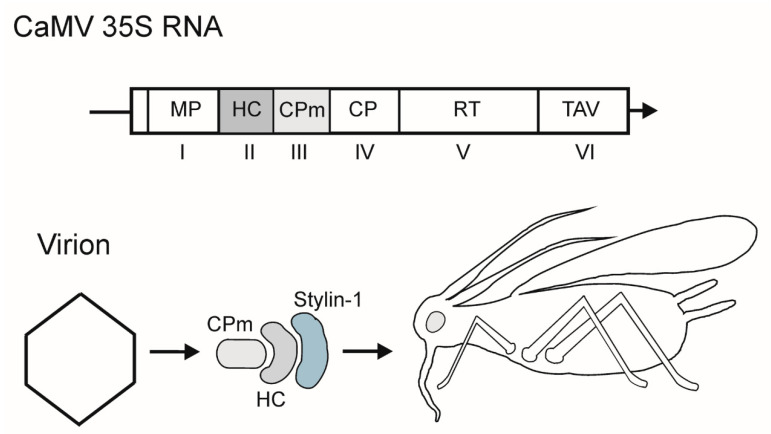
Top, scheme of the 35S mRNA of cauliflower mosaic virus (CaMV). Genes are indicated by Roman numerals. Encoded proteins and protein domains: MP, cell-to-cell movement protein (p1); HC, helper component (p2); CPm, minor capsid protein (p3); CP, major capsid protein; RT, reverse transcriptase; TAV, transactivator/viroplasmine. Arrow indicates the 3′-end of RNA. The genes involved in transmission are shaded. Bottom, scheme of protein interactions in CaMV aphid transmission. Stylin-1, putative receptor in the aphid’s stylet.

**Figure 2 cells-10-00090-f002:**
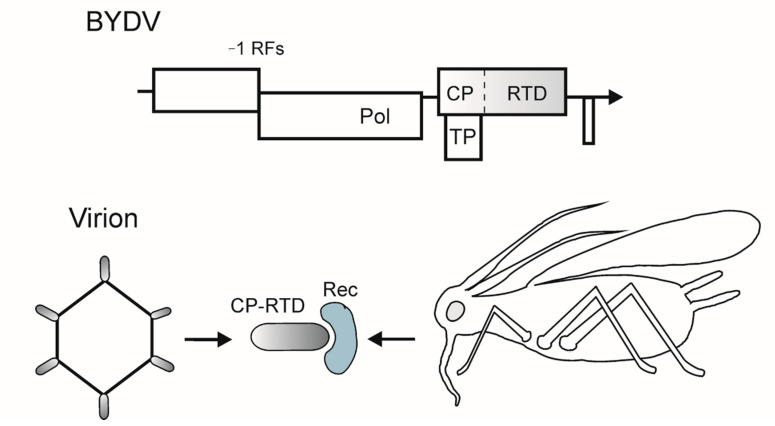
Top, scheme of the genomic RNA of barley yellow dwarf virus (BYDV). −1 RFs, −1 ribosomal frameshifting signal for translation of replication-associated genes. Vertical dotted line, leaky stop codon in the CP gene. Encoded proteins and protein domains: Pol, RNA-dependent RNA polymerase; TP, long distance transport protein; CP, capsid protein; RTD, readthrough domain. The genes involved in vector transmission are shaded. Arrow indicates the 3′-end of RNAs. Bottom, scheme of the protein interactions in luteovirus transmission. The CP-RTD assembles into the particles and interacts with an aphid receptor (Rec).

**Figure 3 cells-10-00090-f003:**
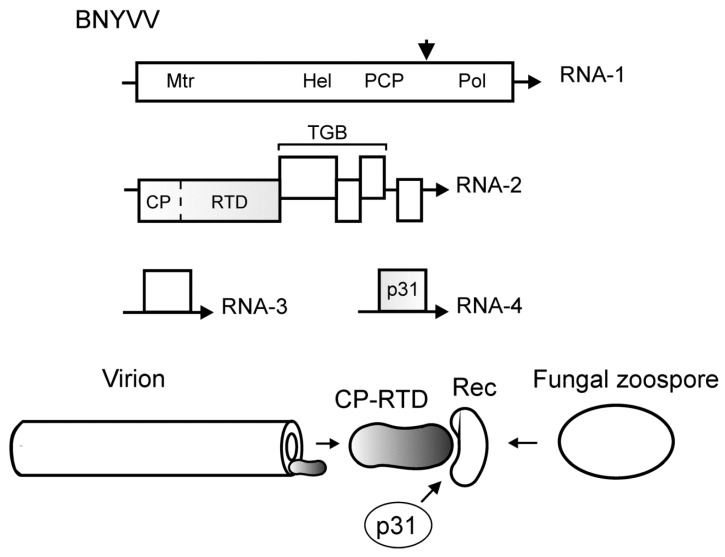
Top, scheme of the RNA genome of beet necrotic yellow vein virus (BNYVV). The genome is divided among RNA-1, RNA-2, RNA-3, and RNA-4. Arrows indicate the 3′-end of RNA. Encoded proteins and protein domains: Mtr, methyltransferase; Hel, helicase; PCP, papain-like cysteine proteinase (arrowhead, the cleavage site); Pol, RNA-dependent RNA polymerase; TGB, triple gene block coding for cell-to-cell movement proteins; CP, capsid protein (dotted line, leaky stop codon); RTD, readthrough domain. The genes involved in transmission are shaded. Bottom, protein interactions in benyvirus transmission by fungal zoospores. CP-RTD is incorporated into one end of the rod-like virion and, with assistance of the BNYVV p31, interacts with a receptor (Rec) in *Polymyxa betae* zoospores to drive the virus transmission.

**Figure 4 cells-10-00090-f004:**
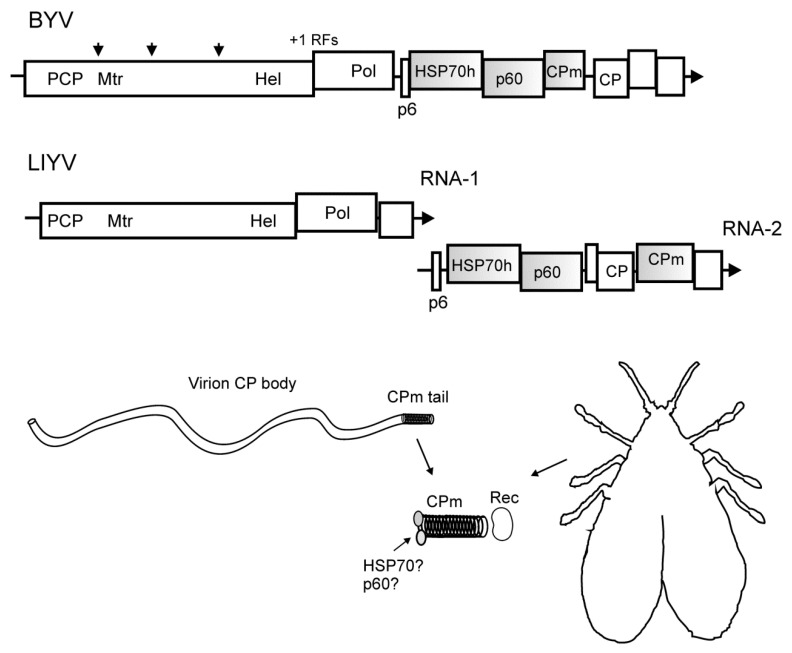
Top, scheme of the RNA genomes of beet yellows virus (BYV, genus *Closterovirus*) and lettuce infectious yellows virus (LIYV, genus *Crinivirus*) drawn approximately to scale. +1 RFs, +1 ribosomal frameshifting signal; arrowheads, cleavage sites in the polyprotein. Abbreviations for encoded protein domains: PLP, papain-like cysteine proteinase; Mtr, N7-guanosine methyltransferase; Hel, RNA helicase; Pol, RNA dependent RNA polymerase; p6, small hydrophobic protein; HSP70h, homolog of the cell HSP70 family of heat shock proteins; p60, ~60 kDa protein; CPm, minor capsid protein; CP, major capsid protein. Arrows indicate the 3′-ends of RNAs. The genes for the capsid proteins are shown as shaded boxes. Bottom, schematic structure of the closterovirus particle with a CPm tail. Localization of the HSP70h and p60 in the virion is shown hypothetically. The CPm, with assistance of HSP70 and p60, interacts with a receptor in aphid (BYV) or whitefly (LIYVV) vector.

## Data Availability

No new data were created or analyzed in this study. Data sharing is not applicable to this article.

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
