# Peer review of "Enhancing Capsid Proteins Capacity in Plant Virus-Vector Interactions and Virus Transmission"

_cells, 2021, doi:10.3390/cells10010090_

Round 1
Reviewer 1 Report
Concerning the title: “Odd” is an odd word: what exactly do you mean? Maybe “unique” or “peculiar” or “special” or “dedicated” might be more appropriate. But this is your choice.
Line 83 reference 19 (Moreno et al. 2005) is for inoculation, Palacios et al. 2002 (PMID: 12466494) might be added for phloem acquisition.
L 92: a better reference than reference 22 is Martinière et al 2013 (PMID 23358702).
L 92 I believe P3 CPm "trimers" is not the correct term? P3 forms triskelia on the capsid, which can be seen as dimers aligning on the capsid surface.
Paragraph starting Line 135: Please verify that you do not mix poleroviruses and luteoviruses.
L 245 this motif is probably for insertion into the capsid.
Figure 2 This is rather a simplified schema of the genome.
Author Response
Concerning the title: “Odd” is an odd word: what exactly do you mean? Maybe “unique” or “peculiar” or “special” or “dedicated” might be more appropriate. But this is your choice.
I agree that "peculiar" and similar terms are more habitual. However, in this case I would like to retain the term "odd", especially when the Referee gives me the choice.
Line 83 reference 19 (Moreno et al. 2005) is for inoculation, Palacios et al. 2002 (PMID: 12466494) might be added for phloem acquisition.
If not absolutely necessary, I would not include the reference to Palacios et al. 2002.
L 92: a better reference than reference 22 is Martinière et al 2013 (PMID 23358702).
I agree. The reference 22 changed as indicated by the Referee.
L 92 I believe P3 CPm "trimers" is not the correct term? P3 forms triskelia on the capsid, which can be seen as dimers aligning on the capsid surface.
Corrected.
Paragraph starting Line 135: Please verify that you do not mix poleroviruses and luteoviruses.
The description follows that of Gray and Gildow 2003 who used the term "luteoviruses". Apparently, they meant "the members of Luteoviridae family" as there seem to be no principal differences in circulative transmission of luteoviruses and poleroviruses. I think, "luteoviruses" may be used in this paragraph.
L 245 this motif is probably for insertion into the capsid.
Exactly, I agree.
Figure 2 This is rather a simplified schema of the genome.
Yes it is, as are the other figures in this review, which are to give the reader an idea of the genome layout , with main focus on the key genes, especially the CP genes.
I wish to thank the Referee for the very useful comments.
Reviewer 2 Report
This review manuscript provides well-organized information on plant virus proteins and vectors.
I would like to quickly read this manuscript with other readers in the journal "Cells".
[Minor points]
please review and organize the references once more.
Line 81, [Moreno et al 2005].?
The B biotype of B. tabaci is being written more as MEAM1 recently. Please write about this word.
Author Response
I wish to thank the Referee for the encouraging abd appreciating comments.
Minor points:
1. The references were checked again and the necessary changes made in the revised version.
2. Line 81 - Moreno et al (2005) - corrected.
3. Biotype names of Bemicis tabaci are as in the paper by Chen et al. 2011. The desigation MEAM1 for Biotype B is added.